American Society for Microbiology | Microbiology Spectrum

# Discrete long-range on-cell motion of bacteriophage T4

Lisa Laura Dreesens,[1] Igor Rutka,[1] Kyriacos Nicolaou,[1] Marie-Eve Aubin-Tam[1]

**ABSTRACT**  The ability of a bacteriophage to distinguish between a suitable and a non-suitable bacterial host is critical for its survival. The series of events occurring between first contact and irreversible binding of a phage to its host is likely playing an essential role in the phage reproduction cycle. However, crucial information about the dynamic interaction between phages and bacterial cell surfaces is still lacking. Until now, most studies have focused on bulk measurements or on analyzing static interactions between phage and host using electron microscopy. These studies generally lack the ability to reveal the spatiotemporal dynamics that are key to understanding the "decision-making" process before the phage commits to infection. Here, we investigated nanoscopic on-cell dynamics using single-particle fluorescence microscopy, which allowed us to track the interaction between fluorescently labeled phage T4 and its host *Escherichia coli* B with high spatial and temporal resolution. We provide the first direct evidence that phage T4 exhibits long-range motion on or near the cell surface, facilitated by repeated shifts in mean positions of its tethering location.

**IMPORTANCE**  We study the interaction of the virus bacteriophage T4 with bacteria. By tagging fluorescent dyes to bacteriophage T4, we can follow the movement of individual viruses when they approach the bacteria, which revealed that these bacteriophages perform a long-range walking motion on the surface of the bacteria.

**KEYWORDS**  bacteriophages, fluorescence assays, spatiotemporal dynamics, on-cell interactions, lipopolysaccharide

Bacteriophages are viruses that infect bacteria to produce progeny, often killing the bacterial host in the process. The first crucial step in the phage's reproduction cycle is to attach to a suitable host. For tailed phages, this attachment is governed by receptor binding proteins (RBPs) present at the end of the tail, that is, tail fibers (TFs), tail spikes or tail tips, which can recognize bacterial host receptors, including porins, lipopolysaccharide (LPS), or teichoic acids (1). Specific interactions between the RBPs and host receptors allow the phage to discriminate between cells and ensure attachment to a suitable host.

Binding kinetic studies suggest that phage attachment to the host occurs in two stages: an initial stage of reversible binding, during which the phage adsorbs to the surface but can still detach, and a second stage of irreversible binding, where the phage commits to infecting the cell (2–5). Further, electron microscopy (EM) studies have revealed that during these stages, conformational changes in the phage's proteins structure facilitate binding and infection (6, 7). These conformational changes are well studied for phage T4 (6, 7), a frequently used model phage with a contractile tail of the family *Myoviridae* that infects *Escherichia* and *Shigella* strains (2, 3). Phage T4 has two sets of TFs, long TFs (LTFs) and short TFs (STFs), which play a role in the binding process. The LTFs reside in a dynamic equilibrium, switching between retracted and extended states, allowing for interaction with a potential host at any given time (6). It is hypothesized that binding of a single LTF would keep the phage tethered to the surface for a short period

Address correspondence to Marie-Eve Aubin-Tam, m.e.aubin-tam@tudelft.nl.

The authors declare no conflict of interest.

See the funding table on p. 14.

of time, allowing a second LTF to bind. Binding of multiple LTFs exerts tension on the phage's baseplate, thereby destabilizing it and releasing the STFs. Binding of at least one STF to a host receptor results in pulling the phage closer to the host surface and fixing the phage in a position that ensures successful infection (6). This added tension leads to further destabilization of the baseplate, causing a cascade of conformational changes within the phage's protein structure (6), marking the switch from reversible to irreversible binding, after which the phage can no longer infect another cell.

Interestingly, across many phage-host pairs, including phage T4 and *Escherichia coli* K12, irreversible binding occurs predominantly at the cellular poles of the host, where receptor concentration is highest (8–11). Moreover, phage binding to its host has been shown to occur with high efficiency (12–14). This raises the question of how phages explore the cell surface to find the target location that facilitates irreversible binding. In a study by Rothenberg et al. (11), interactions between individual fluorescently labeled λ phages and their host were tracked using fluorescence microscopy (FM), thereby revealing that phage λ uses the host receptor-concentration gradient to move over the cell surface. These observations on phage dynamics provide a first insight into how phages localize from a random position on the cell surface to the cellular poles.

The exploration of phage T4 on the cell surface is hypothesized to occur via continuous binding and releasing of the RBP (6, 7). One possible scenario is that individual LTFs from the phage bind and unbind to nearby receptors, causing the phage to reposition in a "tethered walking" manner. Such tethered walking would facilitate movement of the phage over the cell surface toward a more suitable binding location. However, direct observation of these on-cell dynamics for phage T4 is still lacking. Understanding the mechanisms of the targeted search for a suitable host is essential to enable applications such as altering the host range of phages to combat antibiotic-resistant bacteria. To investigate the dynamic interactions between phage T4 and its host in detail, we developed a single-phage fluorescent assay that tracks the location of T4 on host *E. coli* B. We observed the dynamic motion of phage T4 in the vicinity of cells, revealing a discrete step-like behavior, that is, shifts in mean positions of tethering location.

## RESULTS

### Tracking individual T4 phage-host interactions

To investigate the dynamics of T4 phage-host interactions, we adapted a single-particle tracking approach previously described by Rothenberg et al. (11). Fluorescently labeled T4 phages were introduced into a flow cell containing live *E. coli* B cells immobilized on a poly-L-lysine–coated surface. Phage motion was visualized by FM in highly inclined and laminated optical sheet (HILO) mode, while host cells were imaged by phase-contrast microscopy (PCM). Trajectories were reconstructed from time-lapse images using TrackMate (15), enabling analysis of phage behavior at the single-particle level (Fig. 1).

T4 phages were fluorescently labeled by non-specific conjugation of exposed lysine residues to NHS-ester–coupled Alexa dyes (Fig. S1). NHS-ester conjugation is an established method for phage labeling (9, 16, 17). Given the small size of the dyes (Rh = 7.52 Å; MW = 1250 Da) relative to phage T4 and the predominance of labeling sites on the capsid (Fig. S1), steric hindrance during phage-host interactions was expected to be minimal.

We tested whether fluorescent labeling affected phage infectivity. Spotting assays confirmed that labeled T4 phages retained the ability to bind, infect, and propagate. No statistically significant difference in plaque-forming unit (PFU) counts was observed between labeled and unlabeled T4 phage, with the latter being subjected to identical labeling treatment steps without dye ($n = 3$; $P = 0.51$; Fig. S2). Consistent with this result, FM verified adsorption of labeled T4 phages to both *E. coli* B and *E. coli* K-12 host strains (Fig. S3).

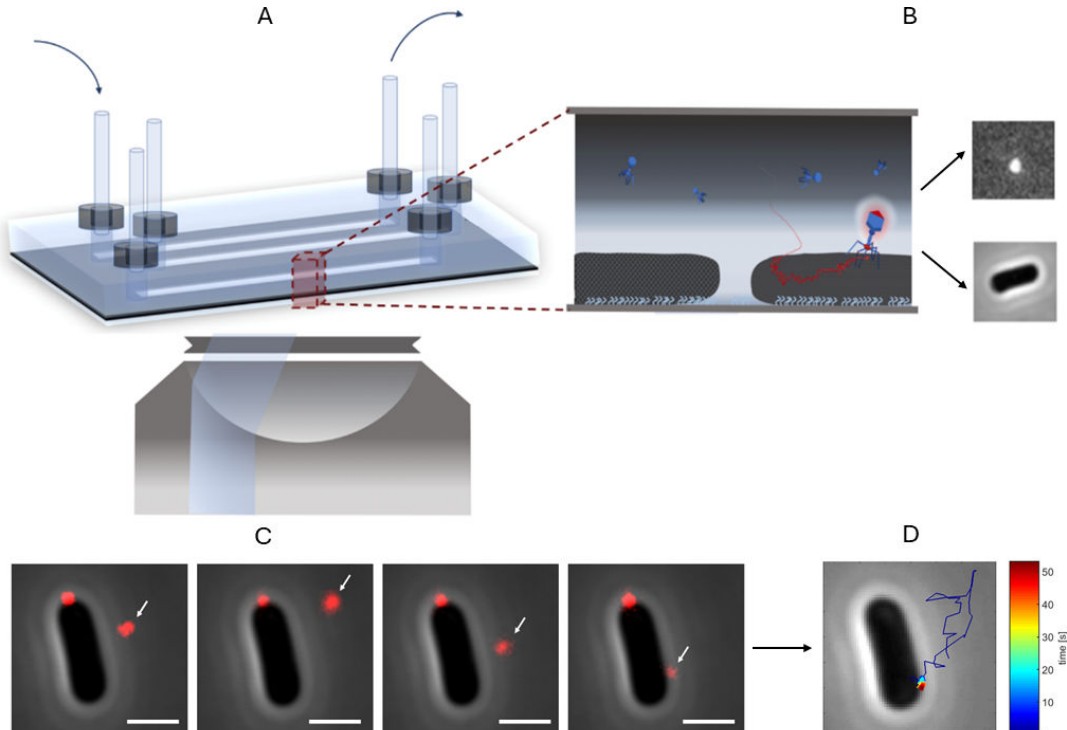

**FIG 1** Experimental set-up for studying phage-host interactions. (A) Schematic representation of the cross-section of the experimental set-up. (B) Fluorescent phages are inserted to one of the three chambers of the flow cell where they can interact with host cells that are immobilized to the poly-L-lysine coated surface. (C) Host cells are visualized by PCM, and fluorescent phages (red foci indicated by an arrow in the image) are tracked over time by fluorescent microscopy in HILO mode. (D) Corresponding phage trajectories are obtained by connecting the corresponding phage positions in each consecutive frame over time using the software package TrackMate (15). Color gradient of obtained exemplary trajectory indicates time, starting in blue and ending in red. Scale bar presents 1 µm.

## Specificity of phage-host binding

To confirm binding specificity, we compared phage adsorption across different host strains. Bulk assays demonstrated irreversible binding of T4 to *E. coli* B via LPS and to *E. coli* K-12 via both OmpC and LPS (Fig. S4) (18). In contrast, T4 showed no irreversible binding with the receptor-deficient *E. coli* K-12 ΔOmpCΔLPS mutant *(*Fig. S4).

Single-particle tracking of at least 300 fluorescently labeled phages in the presence of *E. coli* K-12 ΔOmpCΔLPS supported these findings. No irreversible or prolonged (>2 s) interactions were observed with the receptor-deficient mutant. In *E. coli* B, however, 37% (252/673) of observed trajectories involved cell interactions, and 87% (220/252) of these persisted >2 s (Fig. S5). Although receptor composition differs between *E. coli* B and *E. coli* K-12 and may influence binding kinetics, the absence of detectable interactions with the receptor-deficient mutant indicates that prolonged binding requires specific LTF-receptor interactions.

Together, these results demonstrate that labeling T4 with 10 mM dye produces sufficient signal for reliable detection across the cellular focal plane without impairing infectivity or adsorption specificity. This optimized approach enabled high-resolution tracking of individual phage collisions and interactions with host cells.

## A diverse range of T4 phage-host interaction dynamics

In our assay, we tracked a total of 673 T4 phages at a recording rate of 50 Hz. These trajectories revealed a wide spectrum of motion behaviors. Phages were observed either diffusing freely in solution or interacting with host cells, including (constrained) motion on the cell surface and repeated attachment and detachment events (Fig. 2).

To distinguish the trajectories of phages interacting with a cell, and thus restricted in their motion, from those that are not, we calculated the mean frame-to-frame (FTF)

displacement of each trajectory. The resulting distribution was bimodal: one population centered at 370 nm, corresponding to phages that never co-localized with a cell, that is, *free* phages, and another one below 208 nm, corresponding to phages that (transiently) co-localized with a cell (Fig. 3A).

## Classification of free and interacting phages

We first characterized the motion of *free* phages, that is, trajectories that did not co-localize with a cell (exemplary trajectory in Fig. 2A). Mean squared displacement (MSD) analysis showed a linear relationship with a power law exponent α ≈ 1 (Fig. 4; α = 1.00 ± .21, mean ± SD; $n = 421$), consistent with free Brownian motion (Supplementary 1 and Fig. S6). The calculated diffusion coefficient was 2.56 µm² s⁻¹ (95% CI: 2.53–2.59 µm² s⁻¹; $n = 421$; $R^2 = 0.999$; Fig. S6; [19]).

Next, we focused on the trajectories in the left peak of Fig. 3A, phages that (transiently) co-localized with a cell and displayed lower mean FTF displacement values than *free* phage, indicating that their motion was (temporarily) restricted due to interaction with a cell. We assume that phages irreversibly bound to a cell are restricted in their motion (i.e., bound with multiple LTFs/STFs or in contracted state that cannot cover the surface beyond a circular area with radii of 270 nm—based on a phage tethered to the surface with a single rigid LTF; Fig. S7) and move around their fixed anchoring

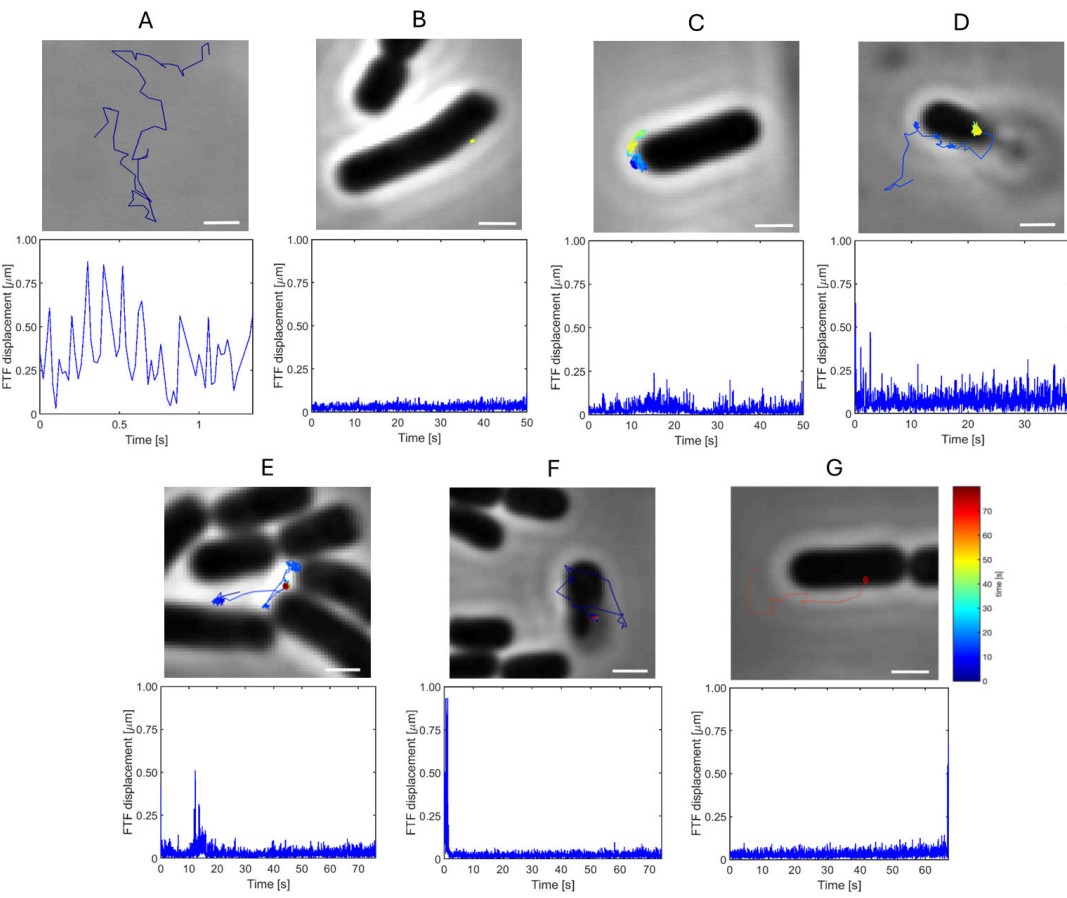

**FIG 2** Phage-host interaction dynamics. T4 phage trajectories were obtained, and net displacement and duration of interaction with the host *E. coli* B were observed. Graphs show the corresponding FTF displacement corresponding to trajectories. Displacement graphs indicate the traveled distance of a phage between consecutive frames as a function of time. Time for each trajectory starts at the timepoint for first detection. (A) phage T4 diffusing freely in bulk (free trajectory), (B) irreversibly bound phage (bound trajectory), (C) interacting phage diffusing in an area beyond the region accessible to a phage tethered with a single LTF (interacting trajectory), (D) phage attaching and detaching multiple times to a single cell (interacting trajectory), (E) phage attaching and detaching multiple times to multiple cells (interacting trajectory), (F) phage transitioning from diffusion in bulk to interaction with a cell (interacting trajectory), and (G) phage detaching from cell surface (interacting trajectory). Color gradient of trajectories indicates time (start: blue; end: red). Scale bars represent 1 µm.

point, resulting in low maximal FTF displacement. In contrast, phages interacting with a cell without being irreversibly bound (i.e., temporarily bound with one or multiple LTFs) should show, at least occasionally, larger FTF displacements than irreversibly bound phages. Therefore, we assessed the maximal FTF displacement of each trajectory that co-localized at least transiently with a cell. The resulting distribution revealed two subpopulations (Fig. 3B). One group displayed very small maximal displacements, consistent with phages anchored at a fixed position on the cell surface (exemplary trajectory in Fig. 2B). These trajectories ($n$ = 112) explored small areas of 104.14 ± 54.35 nm$^2$ (mean area enclosed by convex hull surrounding the trajectory ± SD; Fig. S8) and showed highly restricted sub-diffusive motion (i.e., $a$ < 1; [20]) with $a$ of 0.14 ± 0.09 (mean ± SD; Fig. 4). We categorized these trajectories as *bound*, that is, those presumed to be in or transitioning to the irreversible bound state.

The second group of trajectories ($n$ = 140) exhibited larger displacements consistent with transient tethering to one or more receptors. These trajectories included phages undergoing on-cell displacement (exemplary trajectory in Fig. 2C), repeated attachment and detachment events (exemplary trajectory in Fig. 2D and E), or transitions between free diffusion and cell contact or *vice versa* (exemplary trajectory in Fig. 2F and G). For the latter type of trajectories, we did not observe a gradual increase in FTF displacement prior to detachment events. Closer inspection of the *interacting* phages revealed that 93 of 140 trajectories showed a looser interaction with the cell evidenced by spatial-temporal displacement patterns that featured constrained Brownian motion around a mean point of tethering location combined with less frequent, recurrent relocation of the mean tethering location. From these 93 loosely *interacting* trajectories, 52 phage particles were found capable of exploring a region beyond that accessible from a single tethering point, that is, beyond 270 nm radius (exemplary trajectory in Fig. 5).

## Dynamics of tethered segments

A large number of *interacting* trajectories appeared to contain both intervals in which the phage interacted with the cell, as expected when tethered to LPS receptor(s), as well as intervals in which the phage was undergoing free diffusion. In order to consider these modes (on- and off-cell dynamics) independently in our further analyses, we segmented *interacting* trajectories in intervals during which the phage interacted with the cell (i.e., tethered segments) and intervals during which the phage was free in solution (i.e., off-cell segments). The selection of tethered segments was achieved by setting a conservative threshold. Here, we assumed the following: to account for the skewed distribution of labeling along the phage, we estimate that the center of fluorescence intensity lies around the neck of the phage (i.e. bottom of the capsid;

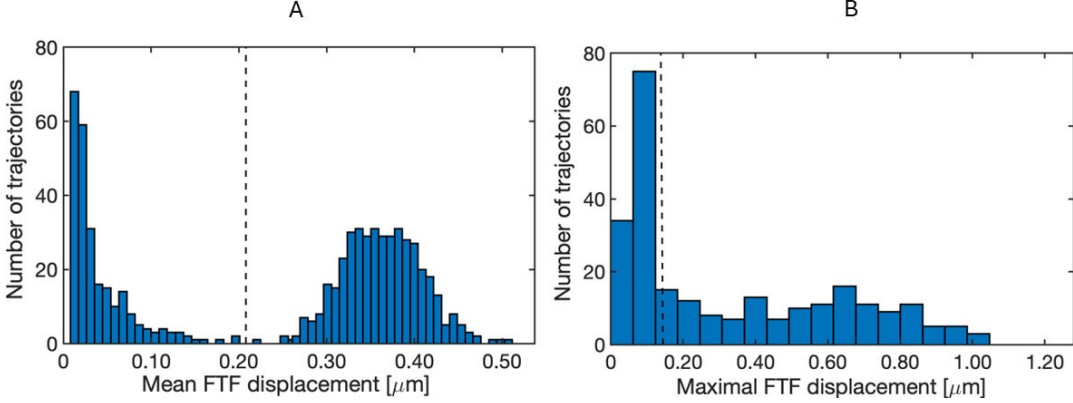

**FIG 3** FTF displacement values of phage trajectories. (A) Histogram showing mean FTF displacement values of all individual trajectories. (B) Histogram showing maximal FTF displacement values for each individual trajectory within the subgroup of trajectories that (transiently) co-localized with a cell. Black dashed line presents the threshold used to separate trajectories that (transiently) co-localized with a cell from those that remain *free* (A), or to separate the *bound* trajectories from *interacting* trajectories (B).

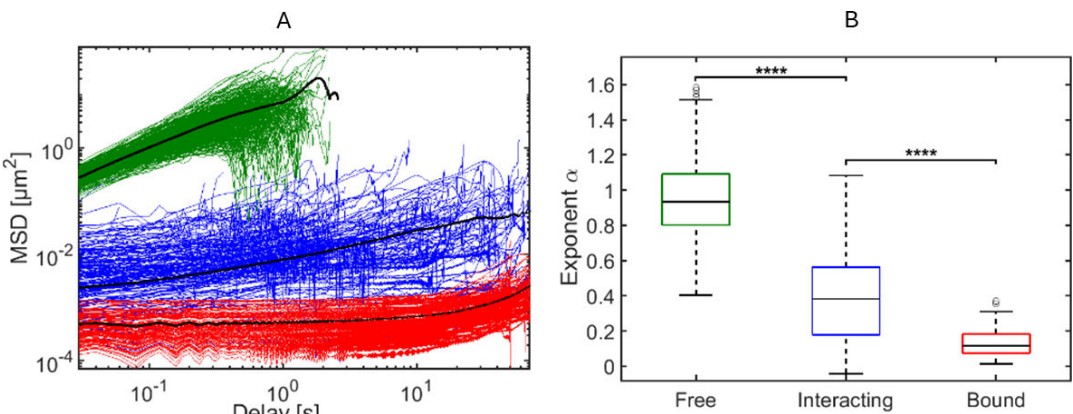

**FIG 4** MSD analysis of *free*, *interacting*, and *bound* phage trajectories. (A) Log-log plot of the calculated MSD as function of delay time for individual T4 phage trajectories assigned to one of three categories: *free* trajectories (green), tethered segments of *interacting* trajectories (blue), and *bound* trajectories (red). Black lines represent mean MSD of each group. (B) Boxplot showing *a* for each category. The box and corresponding black line represent the 25th and 75th percentiles and median, respectively. Error bars indicate the most extreme data points that were within a range of 1.5 times the values of those within the interquartile range (data points within the bottom and top of the box). Data points extending beyond the 1.5 times the interquartile range were marked by black circles (O). ****: *free n* = 421, *interacting n* = 140, and *bound n* = 112; *P* < 0.0001.

Fig. S7A). When such a labeled phage particle is tethered with a single LTF to the surface, its center point is localized in a hemispherical region above the surface with a maximum radius of approximately 270 nm (Fig. S7B). We set a second criterion of a minimum interaction duration of 25 frames (i.e., 0.5 s) to identify the tethered segments. With these criteria, we identified the presumed attachment and detachment points and defined tethered segments. Visual assessment revealed that these tethered segments were almost completely governed by continuous or at least very frequent interactions with the cell. Using this threshold, we identified 210 tethered segments among the 140 trajectories. The mean duration of the tethered segments was 16 ± 22 s (mean ± SD, *n* = 210 segments in 140 trajectories; Fig. S9A). Most tethered segments were short in duration (<3 s), with exceptions that could last the entire duration of the experiment (80 s). MSD-curve analyses of tethered segments revealed a mean *a* of 0.40 ± 0.27 (mean ± SD, *n* = 210 segments). This showed that the phages in these tethered segments

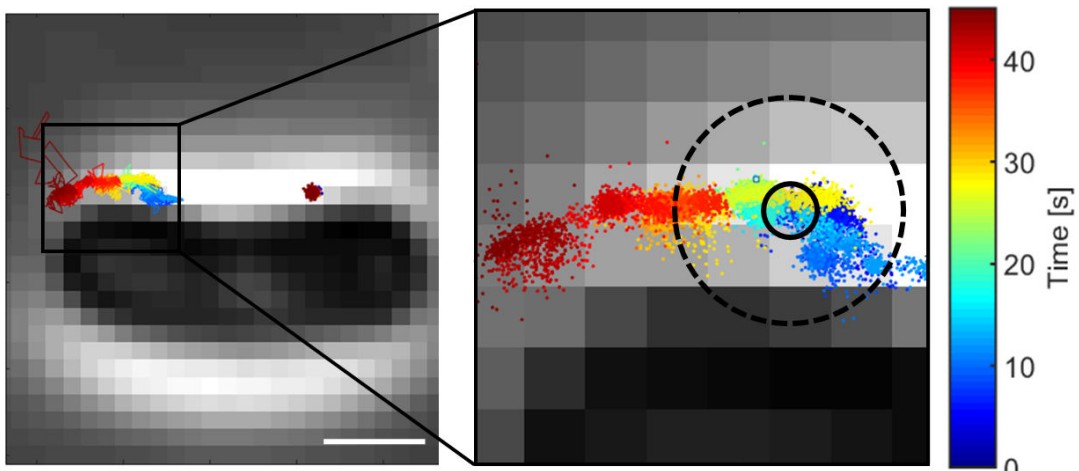

**FIG 5** Characterization of long-range on-cell movement of an interacting phage. An example of *interacting* trajectory is shown at the left side of the cell. A *bound* trajectory can be found on the right side of the cell. On the magnified image of the *interacting* trajectory (right panel), the small black circle corresponds to the interaction area covered by a *bound* phage, and the large dashed black circle shows the maximal theoretical area (radius 270 nm) over which a phage tethered to a single-LTF can move. Scale bar represents 1 μm.

underwent sub-diffusive restricted (anomalous) motion due to phage-cell interactions, albeit to a significantly lesser degree than that observed for *bound* phage trajectories ($n = 271$, $P < 0.0001$; Fig. 4B). Accordingly, the mean area explored by phages in the tethered segments was approximately 10-fold larger (1,198 ± 1,174 nm$^2$; mean ± SD; $n = 210$; Fig. S9B) than that of phages that were tightly bound ($P < 0.0001$).

## Stepwise on-cell motion

To better understand the mechanistic details of the observed long-range on-cell motion, we assessed whether phages moved in a discrete or continued manner. We also recorded trajectories at higher acquisition rates (200 Hz) to increase the temporal resolution of the interaction dynamics. Ten trajectories (six at 50 Hz and four at 200 Hz; Fig. S10) were analyzed in more detail using the Stepfinder algorithm (21, 22). After excluding artifacts such as microscope stage drift, cell movement, and receptor diffusion (Supplementary 2), we observed discrete positional shifts consistent with stepwise exploration of the cell surface.

We analyzed these shifts, by which the phage explores the cell, by fitting steps to the data (21, 22). This allowed us to extract the number of steps (number of shifts in mean position of tethering location), step duration (time for the phage to move from one position to another), stepsize (distance traveled between two mean tethering positions), and dwell time of steps (time the phage spends at a mean tethering position) (Fig. S11). On average, tethered segments contained over 100 steps (Table S1), highlighting the dynamic repositioning of phages during on-cell exploration. The step duration indicated that shifts in tethering position occurred at a rate equal to or faster than the temporal resolution, which was 0.02 or 0.005 s depending on the acquisition rate at which phage trajectories were obtained, reflecting a rapid receptor engagement. A broad range of stepsizes could be observed (Fig. 6A and B), with 42% exceeding 50 nm, above the conservative upper threshold attributable to noise (Supplementary 2). The dwell times of these detected steps ranged from values equal to the acquisition limit (0.02 or 0.005 s) up to almost 10 s (Fig. 6C), indicating stable receptor interactions.

Together, these results reveal that T4 phage-host interactions span a continuum from transient tethering to stable binding. Long-range on-cell motion occurs via discrete, stepwise shifts in tethering position, likely mediated by sequential engagement of LTFs with host receptors. This stepwise mechanism allows phages to explore the cell surface before irreversible adsorption.

## DISCUSSION AND CONCLUSION

Bacteriophages are highly specialized "parasites" capable of discriminating hosts with remarkable specificity. Despite being colloidal particles subjected to Brownian motion, they locate and infect suitable hosts with near-perfect efficiency (2, 3). The critical events bridging first contact and commitment to infection remain poorly understood, as mechanistic insight has largely relied on bulk kinetics or static imaging of interaction configurations (2–4, 6–9). Here, we provide single-particle evidence showing how phage T4 dynamically explores the host surface prior to irreversible adsorption.

Using FM, we show that T4 engages *E. coli* B through repeated attachment-detachment events and sub-diffusive on-cell motion. Behaviors such as reduced diffusivity near or on cells have also been observed for other viruses (23, 24). The dynamics we observed consist of rapid, discrete positional shifts in which the tethered location of T4 repeatedly changes. Importantly, this motion was absent in the receptor-deficient K-12 mutant, supporting that long-range on-cell displacements are mediated by specific interactions between LTFs and host receptors (18, 25). Considering that T4 binding to the LPS of *E. coli* B occurs more frequently via its LTFs than other parts of the phage body (26), and that adhesin (a protein of LTFs that determines the adsorption specificity of T4) binds more strongly to its specific LPS receptor than to non-specific LPSs (27), we conclude that the observed long-range on-cell motion of labeled T4 phage at *E. coli* B results from specific LTF-LPS interaction. We therefore propose that LTF-receptor binding provides

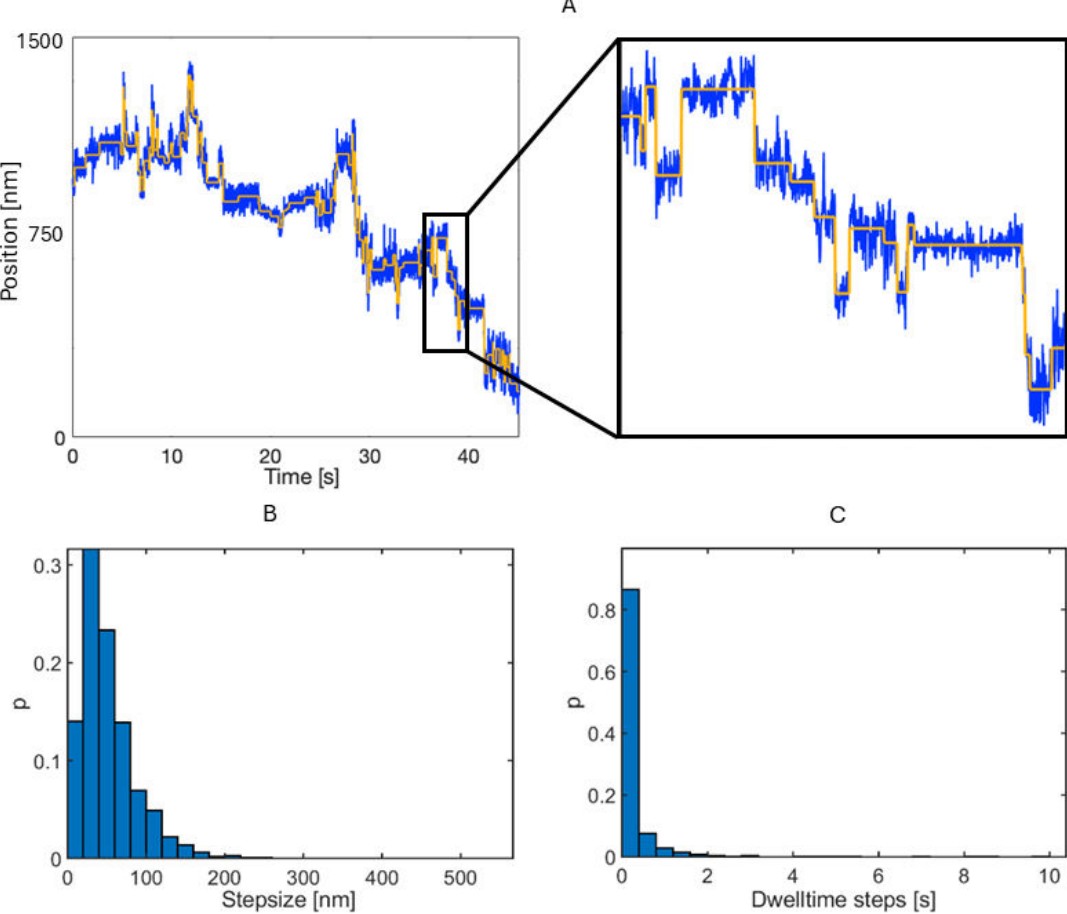

**FIG 6** Characterization of discrete movements in tethered segments of *interacting* trajectories. (A) Position graph of exemplary tethered segment of *interacting* trajectory depicted in Fig. 5, with the blue line presenting the experimentally obtained position in one dimension (*x*-coordinates) as function of time (data obtained at 200 Hz) and yellow line the fit performed by Stepfinder to deduce steps. The proportion of (B) stepsize and (C) dwell time values of steps detected in tethered segments of ten *interacting* trajectories by stepfinder (Fig. S10 and S11).

sufficient interaction strength to tether phages, while maintaining the flexibility needed to scan the cell surface. Hence, this would imply that the LTF-receptor interaction plays an important role in both reversible and irreversible binding. Future studies involving tracking single-particle dynamics of both LTF-deficient T4 phage and LPS-deficient *E. coli* B mutant can examine this hypothesis.

Our findings extend observations made for phage λ (11). Rothenberg et al. (11) reported a decreased diffusion coefficient and the occurrence of spatial focusing on receptor-rich regions near the cellular poles but did not resolve individual repositioning events. In contrast, we observed stepwise motion of T4, involving consecutive shifts in tethered positions, coupled with longer residence times on the cell surface. These differences likely reflect the molecular architecture of T4, which possesses multiple LTFs and targets the abundant receptor LPS ($\sim10^6$ molecules per cell), in contrast to λ, which lacks TFs and binds to the less abundant receptor OmpC ($\sim10^4$ molecules per cell) (28–30). This combination of multiple RBPs and high receptor density likely enables T4 to perform "tethered walking," increasing the chance for a LTF to bind a nearby receptor and keeping it tethered to a local mean position for longer periods. This, in turn, raises the probability of sustained surface residence and efficient progression to irreversible binding.

Our data suggest that T4 acts as a multi-molecular walker, whose on-cell motion may be guided by the receptor distribution and gradient (31), potentially explaining

how phage T4 finds its preferred binding location at the cellular poles (8). Molecular walkers have been reported in other systems, for example, influenza virus (32) and DNAzyme spiders (33–35). Phage T4 may therefore sample the host surface for a suitable binding location through two complementary mechanisms: (i) hopping, where phages attach and detach locally to sample nearby sites, and (ii) LTF-mediated walking, where sequential LTF binding allows exploration over longer distances. Whether phages rely solely on hopping or also use LTF-mediated walking could affect their binding efficiency to the host. The hopping scenario requires a sufficient number of receptors to be present at the contact point in immediate proximity to the phage's LTFs, facilitating a switch to irreversible binding before the phage detaches as a result from an unfavorable binding site. By initially binding with a single LTF and repositioning the LTF to nearby receptors, the phage can move over the surface and explore a larger surface area. Walking reduces dimensionality from 3D diffusion to 2D surface scanning, thereby enhancing the efficiency of locating optimal receptor clusters that promote transition to irreversible binding. This strategy likely improves phage fitness by accelerating the search for an optimal binding location. The idea that phages might increase the target-finding rate by limiting the sampling motion to a fraction of the surface, a concept known as rate enhancement by the reduction of dimensionality (36), was already proposed by Rothenberg et al. for phage λ (11). Testing this model will require experimental modulation of LTF-receptor interactions, for example, by altering receptor density, modifying LTF binding domains, or adjusting L-tryptophan levels known to affect LTF extension (2, 37–40), in parallel with theoretical modeling. This could clarify how phages explore, discriminate, and bind hosts and which factors are most critical.

Not all trajectories involved surface exploration. Some phages adsorbed (almost) irreversibly upon first contact, suggesting that favorable receptor conditions can immediately satisfy binding requirements. Others detached abruptly after prolonged surface confinement, likely reflecting the dissociation of weak or transient bonds or fast-unbinding events beyond the temporal resolution, such as failed injection events disrupting TF-receptor interactions. Although our labeling approach does not directly target the receptor-binding domain (41), steric hindrance from nearby dye attachment cannot be excluded. Future work using alternative labeling strategies, such as intercalating DNA dyes (e.g., SYBR Green), could help rule out such effects.

In conclusion, we provide the first direct evidence that phage T4 exhibits long-range, sub-diffusive motion on the host surface, mediated by discrete, stepwise shifts in tethered positions. The distribution of dwell times at tethered sites provides kinetic information on underlying receptor interactions. This work advances the mechanistic understanding of phage-host engagement and demonstrates that single-particle tracking is a powerful tool to dissect the diverse strategies by which bacteriophages identify and infect their hosts.

## MATERIALS AND METHODS

### Strains, media, and growth conditions

*E. coli* B (ATCC 11303) was used as host indicator strain of phage T4 ATCC 11303-B4. *E. coli* B was grown in lysogeny broth (LB) (Sigma) supplemented with 10 mM $CaCl_2$ and 10 mM $MgSO_4$ at 37°C and 250 rpm to early exponential phase ($OD_{600}$ = 0.1) and infected with T4 at a multiplicity of infection (MOI) 0.5 and grown at 37°C until lysis (clearing of the cell suspension) occurred (~3 h). A final concentration of 2.5% chloroform was added to lyse the remaining cells. The suspension was centrifuged for 20 min at 18,000 *g*, filtered through 0.22 µm filters (SFCA, Minisart NML, Sartorius) and stored at 4°C until further use. The following strains were used for irreversible binding experiments: *E. coli* B (ATCC 11303), *E. coli* K12 (host receptor: OmpC; strain BW25113, genotype rrnB3ΔlacZ4787 hsdR514Δ(araBAD)567Δ(rhaBAD)568 rph-1), and *E. coli* K12 ΔOmpCΔLPS (lacking host receptor; strain TY0728, genotype BW25113 DwaaC::cmDomPC::kan; K12 ΔLPSΔOmpC).

*E. coli* K12 (strain BW25113) and *E. coli* K12 ΔOmpCΔLPS (strain TY0728) were obtained from Washizaki et al. (18).

## Phage purification, activity, and labeling

RNase and DNase (Invitrogen) were added to the phage suspension at a final concentration of 2 U mL$^{-1}$ and incubated for 30 min at RT. Phages were extracted and purified with a CsCl density gradient with the following densities: 1.33, 1.45, 1.50, 1.70, and centrifuged at 28,000 $g$ for 3 h at 4°C, followed by dialysis (7 k cutoff, Thermo Fisher) in MB (20mM Tris-HCl, 100mM NaCl, and 10mM MgSO$_4$) and stored at 4°C until further use.

Impurities and fraction of phage particles with contracted sheath were determined by visualization of negative stain transmission EM (TEM): 3 µL of each sample was placed on a glow-discharged copper EM grid with a 400 Mesh Carbon film (Aurion) for 3 min, followed by three cycles of blotting and washing with 5 µL MilliQ. Each sample was stained for 1 min with 3 µL 1%–2% uranyl acetate followed by one more cycle of blotting and washing. Samples were imaged at JEOL JEM-1400plus TEM, with an acceleration voltage of 120 kV and an energy-dispersive spectroscopy (EDS) detector.

Purified phages were concentrated and washed with an Amicon Ultra-0.5 centrifugal filter unit (100 kDa cutoff, Millipore) in phosphate-buffered saline (PBS) (Sigma) supplemented with freshly prepared 0.1 M sodium bicarbonate. Phages were fluorescently labeled by adding a final concentration of 0.05 mg mL$^{-1}$ Alexa Fluor 488 or 647 NHS Ester in DMSO (Thermo Fisher), unless stated otherwise, following the manufacturer's instructions. The suspension was incubated for at least 3 h at RT. Remaining unbound dye molecules were removed from the fluorescently labeled phage suspension by washing with MB using an Amicon Ultra-0.5 centrifugal filter unit (100 kD, Millipore). The purified labeled phage suspension was stored in the dark at 4°C until further use.

Prior to labeling, phage suspensions were purified by CsCl density gradient centrifugation and quality-checked by TEM (Fig. S12). To identify a dye concentration that allowed for reliable detection, phages were incubated with concentrations ranging from 100 µM to 100 mM. Signal-to-background ratios were qualitatively evaluated from 10 images per concentration using intensity plots (Fig. S13). A concentration of 10 mM yielded sufficient signal-to-background without compromising binding capacity, as confirmed by comparison with a 10-fold higher concentration.

Activity and concentration of the purified labeled phage suspension was confirmed with a spot assay: the sample was serially diluted 10-fold in MB and 10 µL aliquots were placed onto a double-layered agar plate containing the target indicator strain *E. coli* B and supplemented with 10 mM CaCl$_2$ and 10 mM MgSO$_4$. Plates were incubated overnight at 37 °C. Activity was confirmed as a distinct clear "killing"-zone on the bacterial lawn. The concentration of phages in PFU mL$^{-1}$ was determined by counting single plaques visible at spots with higher dilutions.

## Irreversible bulk binding experiments with different *E. coli* strains

Host *E. coli* B, *E. coli* K-12, and *E. coli* K12 ΔOmpCΔLPS were grown in LB to exponential phase (OD$_{600}$ ~ 1.0). A suspension of T4 phages was added to the host at an MOI ~ 0.001 and vortexed shortly, followed by a 15-min incubation period at RT. The phage-cell suspension was run through an Amicon ultra-0.5 centrifugal filter unit (100 kDa cutoff, Millipore) to collect the remaining free phages (i.e., phages that were not bound to cells). The remaining free phage concentration was determined by a spot assay.

## Flow cell

Glass coverslips (22 × 40 mm, No. 1.5, VWR) were extensively cleaned with 2% Hellmanex III (Hellma GmbH). Slides were incubated for 30 min with 0.1% poly-L-lysine (Sigma), washed extensively with MilliQ, and dried with a flow of nitrogen. Holes of approximately 0.1 mm in diameter were drilled in glass slides (76 × 26 × 1 mm, Fisher Scientific) at the beginning and end of each chamber to form an

inlet and an outlet for the sample and buffers. Slides were extensively cleaned with MilliQ and ethanol. Three chambers of the following dimensions: 3.3 cm in length and 0.2 cm in width, were laser-cut in double adhesive tape. The tape served as a spacer between the coated coverslip and glass slide to form a flow cell with three chambers. Press-fit tubing connectors (Grace Bio-Labs) were placed on pre-drilled holes on top of the glass slide to which tubing (0.55 mm inner diameter, 1.07 mm outer diameter, Microbore PRFE Tubing) was connected.

## Infection set-up

For each experiment throughout this study, *E. coli* B cells were grown, from an original stock solution (stored at −80 °C) in LB to early exponential phase ($OD_{600} = 0.3$). Cells were concentrated 10× by centrifuging 1 min at 14,000 g and resuspended in M9 medium (Sigma) in 1/10 of the original volume. Ten microliters of cell suspension was added to the perfusion chamber of the flow cell and incubated on the coverslip for 10 min at RT. Poorly immobilized cells were removed by washing the chamber with 150 μL interaction media at a speed of 175 μL min$^{-1}$ using a syringe dispenser. A suspension of labeled T4 phages was added to the perfusion chamber containing immobilized *E. coli* B cells. The dynamics between individual phages and cells were imaged by FM and PCM.

## Imaging conditions

Image acquisition at low frame rate (50 fps) was executed on the Olympus inverted microscope (IX81) with a TIRF illumination system. Phage trajectories were imaged by FM via fluorescence excitation of fluorophore Alexa480, provided by an Olympus CW laser at 488 nm with a maximum output power of 150 mW. As detection device, an Electron Multiplying CCD (Andor iXon X3 DU897) containing a chip size of 512 × 512 pixels with a resolution of 80 nm pixel$^{-1}$ was used. The electron-multiplying gain was set to 200. A 0.6 ND filter was used to reduce overexposure. Emission filter 524/29 was used. Imaging was performed in HILO mode. A UPlanFLN objective was used with magnification 100×, numerical aperture 1.3, and working distance 0.2 mm. A total of 500 images with PCM and 2,000 or 4,000 images with FM were taken with an exposure time of 100 and 20 ms, respectively. The laser intensity was set at 100–200 *mW*.

Image acquisition at a high frame rate (200 fps) was obtained on a Nikon inverted microscope (Ti2-E) with a TIRF illumination system (Gataca iLAS2). Images of 200 nm TetraSpeck beads (Invitrogen) were taken both with FM and PCM. Phage trajectories were imaged by FM via fluorescence excitation of fluorophore Alexa480, provided by an Olympus CW laser at 488 nm with a maximum output power of 150 mW. The detection device was a dual set-up with an CCD (Andor iXon Ultra 897) with a chip size of 1,004 × 1,025 pixels for FM and an Electron Multiplying CCD (Retiga R1) with a chip size of 1,376–1,024 pixels for PCM with a resolution of 161 and 63.5 nm pixel$^{-1}$, respectively. A Nikon Apo TIRF objective was used with magnification 100×, numerical aperture 1.49, and working distance 0.12 mm. Emission filter 525/50 was used. FM imaging was performed in HILO mode. To obtain phage trajectories, 2,000 images were taken at an exposure time of 4 ms and a laser intensity of 198 *mW*. *E. coli* B cells were imaged with PCM by taking 200 images with an exposure time of 100 ms.

To confirm detection of labeled phages across the full height of a bacterial cell, we performed z-stack imaging in 200-nm increments. Labeled phages bound to glass and cell surfaces remained detectable up to 1.5 μm from the focal plane, ensuring reliable detection of phage-cell interactions throughout the cellular volume (Fig. S14).

## Data processing and statistics

### Alignment of fluorescent and phase contrast images taken at high frame rate (200 Hz)

Obtained images of TetraSpeck beads (Invitrogen) were used to determine the alignment coordinates for correct overlay between images obtained by PCM (*E. coli* B cells) and FM (labeled T4 phage). Background subtraction and inversion were applied on bead images taken with PCM, followed by downscaling of the image to 0.39 of its original size. Alignment coordinates for overlaying the images obtained by PCM with those obtained by FM were obtained by the descriptor-based registration (2d/3d) plugin of Fiji (42). Coordinates were used for alignment of all pre-processed (downscaled) phase contrast images of *E. coli* B cells, followed by cropping the PCM images to the same size as the FM image.

### Determination of cell outline

The outline of cells imaged by PCM was determined by enhancing the contrast in ImageJ by saturating 0.03% of pixels, followed by applying a Gaussian blur filter with sigma 2, and a threshold by Yen method (43). Using Matlab, *E. coli* B cells were detected as binary objects if a minimal area of 100 and maximal area of 1,000 pixels was detected, with a minimal eccentricity of 0.7 and minimal solidity of 0.9.

### Cell movement analysis

A stack summation was performed on binarized images of cell outlines using ImageJ. This allowed for the identification of rotational movement of the cell or translational movement due to drift in image plane caused by the microscope stage, with pixel-level accuracy. The rotational or translational movements were visible as an imperfect overlap of individual pixels within binary objects throughout the movie, and such identified cells with corresponding trajectories were excluded from further analysis. For a subset of cells, an in-depth cell movement analysis was performed with the method described in Supplementary 2.

### Phage trajectories

Sequences of images were used to obtain phage trajectories using the algorithm TrackMate (15) in ImageJ. Here, a Laplacian of Gaussian filter was used to detect the center position of each fluorescent particle (phage) within a two-dimensional plane. The center position within each consecutive frame (i.e., image) was linked over time using the built-in linking tool with a threshold of 1,200 nm for the linking max distance, gap-closing max distance, and gap-closing max frame gap resulting in a phage trajectory. Tracks containing <25 points within a trajectory were discarded. Downstream analysis of trajectories was performed with Matlab using @msdanalyzer (19). Trajectories obtained at the image edge were automatically discarded using the function edgeDiscard. Trajectories that overlapped with a minimum of three other tracks at the same pixel were automatically discarded using the function overlapDiscard. Trajectories interacting with the glass substrate, present on moving or poorly defined bacteria, or artifacts due to erroneous linking of particles were manually discarded. α-Values were determined by fitting a line to the first 25% of each individual phage trajectory. For the *interacting* trajectories, only on-cell segments with a duration of ≥0.5 s were included. The diffusion coefficient was calculated through a linear weighted fit of the mean MSD curve. Classification of the trajectories into two categories, "interacting with cell" and "free", was achieved by placing a threshold at 208 nm which separated the two classes with 99% accuracy based on visual inspection. Some trajectories briefly co-localized with a cell at the start or end of the captured trajectory and, as a result, had mean displacement values slightly larger than 208 nm. These were manually transferred to the "interacting with cell" category. Classification of *bound* and *interacting* trajectories was realized by

placing a threshold at a maximal FTF displacement value of 144 nm. Misclassification of trajectories into two subgroups occurred around the set threshold value (approximately 1% of trajectories) when trajectories started almost directly after attachment, stopped directly after detachment with the cell (≥ 1 frame in which the phage was free in diffusion), or showed very slow movement of position in time. Therefore, the trajectory path of each trajectory was visually inspected and manually transferred to the other group when needed. Qualitative visual assessment checked (i) co-localization of the trajectory paths with the cell during the movie, (ii) change of apparent position in time (shift of mean position in time, no circular pattern), and (iii) covering an interaction area beyond 270 nm radii. Start and endpoint of on-cell segments of *interacting tethered* trajectories were identified when the FTF displacement remained below 270 nm for at least 25 consecutive frames.

### Motion analysis on-cell segments phage trajectories

Discrete stepwise motion within on-cell segments of phage trajectory was analyzed using the algorithm Stepfinder, as described by Kerssemakers et al. (22) and Loef et al. (21). Here, the number of iterations was set at 3 1/4 of the number of frames within a movie, and the acceptance threshold was set at 0.15 as described by Kerssemakers et al. (22).

### Statistics

Significance was calculated using the Mann-Whitney $U$ test, where a significant difference was assumed when $P < 0.05$. Whenever needed, a Bonferroni correction was applied.

### Experimental independence

Data obtained in this study are built from two data sets. The main data set consists of trajectories obtained at an imaging rate of 50 Hz, while additional data, used to study the details of on-cell interactions with higher temporal resolution, was obtained at an imaging rate of 200 Hz. Trajectories imaged at 50 Hz were collected in three experiments (i.e., three flow cell chambers) all on the same day, while trajectories imaged at 200 Hz were collected in one experiment (i.e., one flow cell chamber) on a different day and with a different microscope set-up from those obtained at an imaging rate of 50 Hz. Both data sets made use of a bacterial cell culture that was grown from the same original stock solution. For each data set, we made use of a batch of labeled phages (i.e., two batches) that was grown from the same original phage stock.

### ACKNOWLEDGMENTS

The authors thank Bertus Beaumont for his help with designing and planning the experiments, with data analysis, and with preparing the manuscript. The authors are grateful for helpful discussions with Wouter Liefting and Jacob Kerssemakers on the usage of the Stepfinder algorithm and with Jérémie Capoulade for the microscope set-up. The authors thank Cleo Bagchus for helping with optimizing the adhesion of cells to the flow cell and Lars Houtman for the help in the lab obtaining data for the binding efficiency bulk experiment with mutant cells and Richard Sportsman for reviewing the manuscript.

### AUTHOR AFFILIATION

[1]Department of Bionanoscience, Kavli Institute of Nanoscience, Delft University of Technology, Delft, the Netherlands

## AUTHOR ORCIDs

Marie-Eve Aubin-Tam ⓘ http://orcid.org/0000-0001-9995-2623

## FUNDING

| Funder | Grant(s) | Author(s) |
|---|---|---|
| Nederlandse Organisatie voor Wetenschappelijk Onderzoek | | Marie-Eve Aubin-Tam |

## AUTHOR CONTRIBUTIONS

Lisa Laura Dreesens, Data curation, Formal analysis, Investigation, Methodology, Project administration, Writing – original draft, Writing – review and editing | Igor Rutka, Methodology, Writing – review and editing | Kyriacos Nicolaou, Methodology, Writing – review and editing | Marie-Eve Aubin-Tam, Conceptualization, Funding acquisition, Investigation, Project administration, Supervision, Writing – review and editing

## DATA AVAILABILITY

The data that support the findings of this study are openly available in 4TU. Research-Data repository at
https://doi.org/10.4121/0a47bd92-9b92-4564-9a75-2f83d60e646f

## ADDITIONAL FILES

The following material is available online.

### Supplemental Material

**Supplemental material (Spectrum02509-25-S0001.pdf).** Fig. S1 to S14; Table S1.

### Open Peer Review

**PEER REVIEW HISTORY (review-history.pdf).** An accounting of the reviewer comments and feedback.

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
