## [Reviewer comments · Microbiology Spectrum]

Microbiology Spectrum

Discrete long-range on-cell motion of bacteriophage T4

Lisa Dreesens, Igor Rutka, Kyriacos Nicolaou, and Marie-Eve Aubin-Tam

Corresponding Author(s): Marie-Eve Aubin-Tam, Technische Universiteit Delft

Review Timeline:

Submission Date:	September 17, 2025
Editorial Decision:	October 3, 2025
Revision Received:	November 13, 2025
Accepted:	November 19, 2025

Editor: Hui Wang

Reviewer(s): The reviewers have opted to remain anonymous.

Transaction Report:

DOI: <https://doi.org/10.1128/spectrum.02509-25>

Re: Spectrum02509-25 (Discrete long-range on-cell motion of bacteriophage T4)

Dear Dr. Marie-Eve Aubin-Tam:

I am pleased to inform you that your manuscript has been editorially accepted for publication. However, there were concerns that the English language usage in the manuscript might make it difficult to properly evaluate the science. I recommend that you ask a colleague of yours who is a native English speaker to read and provide you some feedback on the writing. You are also welcome to use one of the services here: <https://journals.asm.org/writing-your-paper#language-editing-services>. The use of these services will have no direct bearing on the editorial decision. ASM has no affiliation with these companies. Also, there are a few additional questions in the submission form that need to be answered before the final decision. Once these are completed, please return your submission so that I can move your paper forward to acceptance.

Thank you for transferring your paper to Spectrum.

Sincerely,
Hui Wang
Senior Editor
Microbiology Spectrum

Re: Spectrum02509-25R1 (Discrete long-range on-cell motion of bacteriophage T4)

Dear Dr. Marie-Eve Aubin-Tam:

Your manuscript has been accepted, and I am forwarding it to the ASM production staff for publication. Your paper will first be checked to make sure all elements meet the technical requirements. ASM staff will contact you if anything needs to be revised before copyediting and production can begin. Otherwise, you will be notified when your proofs are ready to be viewed.

Sincerely,
Hui Wang
Editor
Microbiology Spectrum